# Ranking lifestyle risk factors for cervical cancer among Black women: A case-control study from Johannesburg, South Africa

Mwiza Gideon Singini[1,2], Freddy Sitas[3,4,5], Debbie Bradshaw[3], Wenlong Carl Chen[1,6], Melitah Motlhale[1,2], Abram Bunya Kamiza[6], Chantal Babb de Villiers[7], Cathryn M. Lewis[8,9], Christopher G. Mathew[6,9], Tim Waterboer[10], Robert Newton[11,12], Mazvita Muchengeti[1,2,13], Elvira Singh[1,2]*

1 National Cancer Registry, National Health Laboratory Service, Johannesburg, South Africa, 2 Division of Epidemiology and Biostatistics, School of Public Health, Faculty of Health Sciences, University of Witwatersrand, Johannesburg, South Africa, 3 Burden of Disease Research Unit, South African Medical Research Council, Cape Town, South Africa, 4 Centre for Primary Health Care and Equity, School of Public Health and Community Medicine, University of New South Wales Sydney, Australia, 5 Menzies Centre of Health Policy, School of Public Health, University of Sydney, Australia, 6 Sydney Brenner Institute for Molecular Bioscience, Faculty of Health Sciences, University of the Witwatersrand, Johannesburg, South Africa, 7 Division of Human Genetics, School of Pathology, Faculty of Health Sciences, University of the Witwatersrand, Johannesburg, South Africa, 8 Social, Genetic and Developmental Psychiatry Centre, Institute of Psychiatry, Psychology & Neuroscience, King's College London, London, United Kingdom, 9 Department of Medical and Molecular Genetics, Faculty of Life Sciences and Medicine, King's College, London, United Kingdom, 10 Division of Infections and Cancer Epidemiology, German Cancer Research Center (DKFZ), Heidelberg, Germany, 11 MRC/UVRI and LSHTM Uganda Research Unit, Entebbe, Uganda, 12 University of York, York, United Kingdom, 13 South African DSI-NRF Centre of Excellence in Epidemiological Modelling and Analysis (SACEMA), Stellenbosch University, Stellenbosch, South Africa

* elviras@nicd.ac.za

## Abstract

### Background

Aside from human papillomavirus (HPV), the role of other risk factors in cervical cancer such as age, education, parity, sexual partners, smoking and human immunodeficiency virus (HIV) have been described but never ranked in order of priority. We evaluated the contribution of several known lifestyle co-risk factors for cervical cancer among black South African women.

### Methods

We used participant data from the Johannesburg Cancer Study, a case-control study of women recruited mainly at Charlotte Maxeke Johannesburg Academic Hospital between 1995 and 2016. A total of 3,450 women in the study had invasive cervical cancers, 95% of which were squamous cell carcinoma. Controls were 5,709 women with cancers unrelated to exposures of interest. Unconditional logistic regression models were used to calculate adjusted odds ratios ($OR_{adj}$) and 95% confidence intervals (CI). We ranked these risk factors by their population attributable fractions (PAF), which take the local prevalence of exposure among the cases and risk into account.

Committee approval. Data are available from the SA-NCR /National Health Laboratory Services. Contact adrianp@nicd.ac.za for researchers who meet the relevant ethics criteria for access to these data.

**Funding:** This study was supported by the South African Medical Research Council (with funds received from the South African National Department of Health) and the UK Medical Research Council (with funds from the UK Government's Newton Fund) (MRC-RFA-SHIP 01-2015). The funders were not involved in the conceptualisation, review or approval of the manuscript. This project forms part of an international research program aimed at identifying evolving risk factors for cancer in African populations (ERICA-SA). (https://www.samrc.ac.za/intramural-research-units/evolving-risk-factors-cancers-african-populations-erica-sa).The funders had no role in study design, data collection and analysis, decision to publish, or preparation of the manuscript.

**Competing interests:** The authors have declared that no competing interests exist.

## Results

Cervical cancer in decreasing order of priority was associated with (1) being HIV positive ($OR_{adj}$ = 2.83, 95% CI = 2.53–3.14, PAF = 17.6%), (2) lower educational attainment ($OR_{adj}$ = 1.60, 95% CI = 1.44–1.77, PAF = 16.2%), (3) higher parity (3+ children vs 2–1 children ($OR_{adj}$ = 1.25, 95% CI = 1.07–1.46, PAF = 12.6%), (4) hormonal contraceptive use ($OR_{adj}$ = 1.48, 95% CI = 1.24–1.77, PAF = 8.9%), (5) heavy alcohol consumption ($OR_{adj}$ = 1.44, 95% CI = 1.15–1.81, PAF = 5.6%), (6) current smoking ($OR_{adj}$ = 1.64, 95% CI = 1.41–1.91, PAF = 5.1%), and (7) rural residence ($OR_{adj}$ = 1.60, 95% CI = 1.44–1.77, PAF = 4.4%).

## Conclusion

This rank order of risks could be used to target educational messaging and appropriate interventions for cervical cancer prevention in South African women.

## Introduction

Cervical cancer is the fourth most frequently occurring cancer type and the fourth leading cause of death from cancer among women globally [1]. In sub-Saharan Africa (SSA), more than 101,423 new cases and 76,444 deaths of cervical cancer occur each year [1]. In South Africa, the age-standardised incidence rate of cervical cancer as reported by Globocan is 44.4 per 100,000 women per year [1]. According to the 2017 South African National Cancer Registry report, there were 5,630 new histologically confirmed cases of cervical cancer amongst black women with a lifetime risk (0–74 years) of 1 in 33 women [2].

The main risk factor for cervical cancer and its premalignant lesions is persistent infection with high-risk Human papillomaviruses (hr-HPV) [3]. A previous study from the Johannesburg Cancer Study (JCS), showed a seroprevalence of 78% for anti HPV-16 antibodies in cervical cancer cases [4]. However, at least 90% of women with hr-HPV infection, do not develop cervical cancer [3]. This suggests that other environmental cofactors acting jointly with hr-HPV elevate the risk of cervical carcinogenesis [5].

Epidemiological studies have indicated human immunodeficiency virus (HIV), parity, smoking, alcohol consumption, contraceptive use, age, education and number of sexual partners to be cofactors of cervical cancer pathogenesis [5–12], but these cofactors vary in prevalence (and possibly the risk) in different settings. Understanding the prevalence levels of different cofactors and their risk for cervical cancer in a local setting may help in identifying risk profiles to target in cervical cancer prevention.

In South Africa, studies on some lifestyle-related cofactors for cervical cancer were published about 10 years ago from the JCS [13–15]. These studies focused separately on individual cofactors such as HIV [15], smoking [14] or contraceptive use [13]. Among black South African women little is known about the relative importance of these cofactors for cervical cancer.

In South Africa, the JCS was established in 1995 to redress at that time the historical paucity of epidemiological cancer research among black South Africans [16]. The JCS aimed to examine whether key known and emerging risk factors for cancer in women of mainly European ancestry applied to patients of African ancestry in Johannesburg, South Africa. These aims evolved into measuring the importance of known and emerging risk factors for cancer in a local setting.

We analysed JCS lifestyle data to evaluate the contribution of different cofactors in the pathogenesis of cervical cancer among black South African women. In the current study, we used a

subset of the JCS female samples and focused on ages 25 to 64 years to align with current cervical screening guidelines but also allowing for a more comparable assessment of the contribution of different cofactors in rank order to the risk of cervical cancer.

## Methods

### Setting and participants

The details of the JCS have been described elsewhere [16], but briefly, the JCS recruited over 26,000 black South African patients (both sexes) who were newly diagnosed with an incident cancer between 1995 and 2016. Recruitment took place mainly at Charlotte Maxeke-Johannesburg Academic Hospital medical oncology and radiotherapy clinics and associated peripheral clinics. Trained interviewers collected self-reported data on demographics and key lifestyle risk factors from consenting participants using a structured questionnaire, and took peripheral blood samples for HIV and other analyses. The questionnaire included questions on the following: socio-demographic factors such as; place of birth and residence, marital status, education, the home language of parents. Enviromental exposures such as method of cooking and heating. Lifestyle factors such as; smoking by type of tobacco and amounts smoked, snuff (sniffed tobacco) use, alcohol consumption by type, parity, use of oral and injectable contraceptives, number of sexual partners. On occupations, self-reported use of Anti-Retroviral Therapy (ART) (since 2005), PAP smear (2001) and self-reported history of diabetes. The JCS and the current study were approved by the University of the Witwatersrand Human Research Ethics Committee (Medical) (certificate number for the current study. M200252). In the JCS, participants gave written informed or wittnessd consent to once-off interview and optional blood draw and to have their information and blood sample anonymized. Any future investigations require approval of the University of the Witwatersrand Human Research Ethics Committee HREC [16].

### Selection of cases and controls

The JCS is amenable to a case-control design and analysis. Cases were women who were newly diagnosed with invasive cervical cancer (histologically confirmed squamous cell carcinoma and adenocarcinoma) and controls were designated as women with newly diagnosed cancers that have no known relationship with the exposures of interest [16].

A total of, 26,263 participants were enrolled in the JCS between 1995 to 2016. We excluded 8,677 (33.0%) males, 3,105 (17.6%) women older than 65 years or younger than 25 years, 663 (4.6%) non-cancer participants, 945(6.9%) non-South Africans, 1 with missing data, 640 (5.0%) with missing HIV-status and 691 (5.7%) with primary site unknown malignancy. From those with cancer of the cervix, we excluded International Classification of Diseases for Oncology (ICD-O) codes such as (ICD-O morphology: 8010–8050 epithelial neoplasm (n = 98, 2.6%), (ICD-O morphology: 8000 and 8001) not otherwise specified (n = 9, 0.2%), (ICD-O morphology: 8560, 8570 and 8574) complex epithelial neoplasms (n = 108, 2.8%) and other minor histological types (ICD-O morphology: 8098,8170, 8200, 8272, 8310, 8441,8460, 8480–8490, 8500,8650 and 8931–9100) (n = 64, 1.7%) Fig 1.

Fig 1 also outlines the criteria used to select cases and controls. The control groups were arranged into four sets. Each set comprised of women diagnosed with different cancer types that are unrelated to the exposures of interest (these being infection, smoking, alcohol, parity, number of sexual partners and use of hormonal contraceptives) [16–18] Fig 1. Cancer controls unrelated to infection constituted the highest number of controls S1 Table. The demographic analyses excluded 2,009 (26.0%) women with cancers related to infections. Analyses relating to smoking excluded 2,218 (28.7%) women with cancers related to smoking and infections.

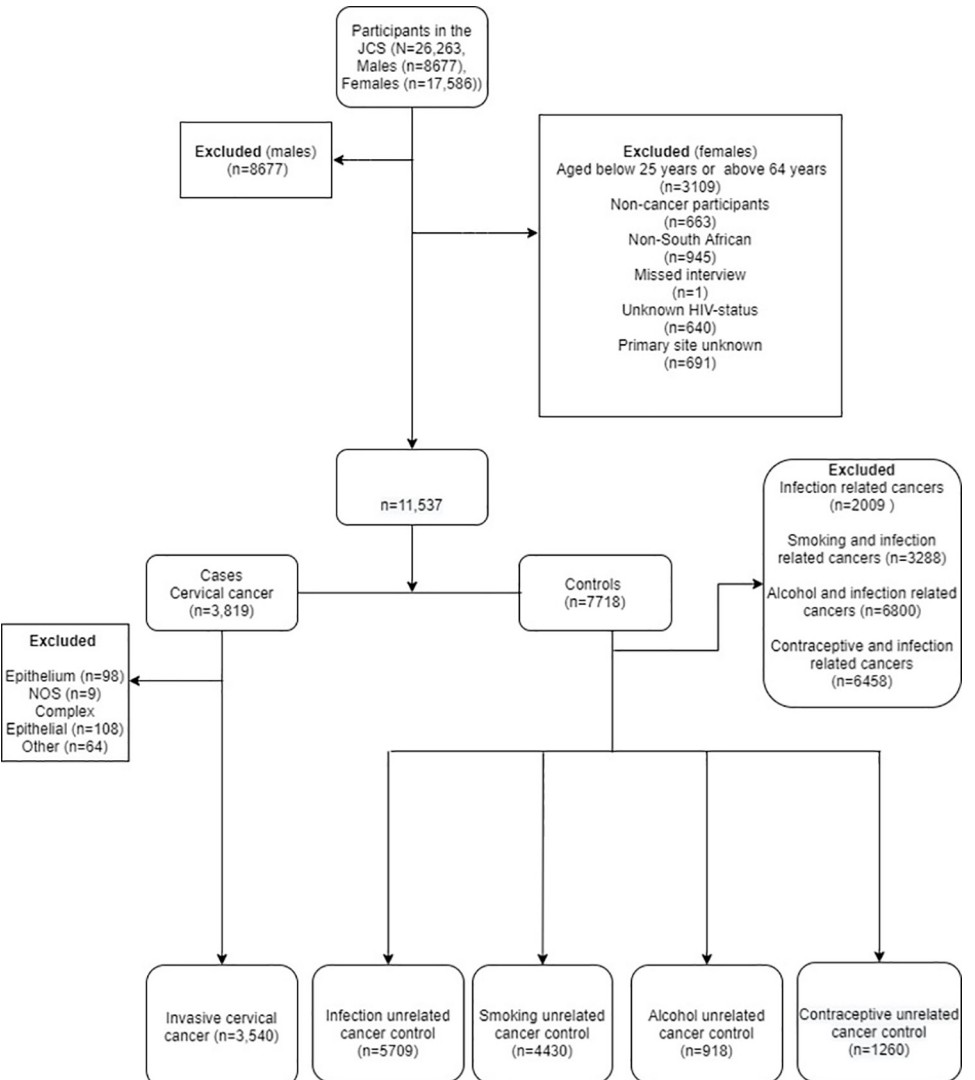

**Fig 1. Data flow diagram on the selection exposure unrelated cancer controls.**

Analyses relating to alcohol excluded 6,800 (60.4%) women with cancers related to alcohol and infections. Analyses relating to hormonal contraceptives excluded 6,458 (57.4%) women with cancers related to hormonal contraceptives and infections. Of the remaining 9,249 women, 3,540 (38.3%) had invasive cervical cancer (defined as cases) and 5,709 (61.7%) had other cancers (defined as controls) that were not related to the exposure of interest Fig 1.

## Definition of exposures

For other demographic factors, women were grouped according to the period of the interview (1995–1999, 2000–2004, 2005–2009, 2010–2016), education level (none, primary, secondary and tertiary and unknown), marital status (never married, married, ever married and unknown) and place of residence (rural, urban and unknown).

Women's HIV status (positive and negative) was based on Abbott Axysm HIV1/2 gO Microparticle Enzyme-linked Immunoassay (ELISA) (1995 to 2005) and the Vironostika (HIV UniForm II plus O) micro ELISA (2006 to 2016) test results [15], while the other cofactors

were self-reported. Use of hormonal contraceptive was categorized into (never (injectable and oral), ever oral/ not injectable, ever injectable/ not oral, ever oral, injectable and ever injectable and/ or oral) and unknown [13]. Parity was categorized into 1–2, > = 3 children and unknown. We excluded 168 (1.9%) women who had never given birth (on assumption that some may have had pre-existing unreported gynaecological conditions or hysterectomies). The number of sexual partners for the participants were grouped into 0–1, 2–5, > = 6 and unknown.

Women's smoking status was classified into never (non-smokers), ex-smoker, current smoker and unknown. The current smokers included those women who reported to have quit smoking within 5 years of interview (diagnosis) to reduce the likelihood of reverse causation [14]. We calculated total alcohol consumption based on different types of alcoholic beverages consumed: spirits (homebrew and commercial), beer (homebrew and commercial), wine, maize and sorghum, the content of ethanol from each type of alcohol beverages and the number of drinks per day. The average number of drinks per week was used to calculate the total alcohol consumption [19]. We defined one drink as to have drunk 15.7 grams of unmixed alcohol beverage, which is equivalent to one bottle of beer of 340ml that contains 4% of ethanol. Based on the Center for Disease Control (CDC) definitions [20], we categorized women's self-reported consumption of alcoholic beverages into never, light (31.4 grams or less of alcohol per week) and heavy (47.1 grams or more of alcohol per week).

## Statistical analysis

Demographic characteristics were summarized using frequencies and percentages for the categorical variables, medians and the interquartile range (IQR) for the continuous variable that was not normally distributed. We then analyzed the relationship between each of the six cofactors with cervical cancer, using the appropriate set of cancer controls for each analysis Fig 1.

We assessed for HIV status in model 1, parity in model 2, smoking in model 3, alcohol consumption in model 4; hormonal contraceptive use in model 5, and the number of sexual partners in model 6. We fitted our data using a backward stepwise multivariable unconditional logistic regression to estimate the unadjusted and adjusted odds ratios ($OR_{adj}$) and 95% confidence intervals (CI). In the final multivariable analysis of each model, we returned to the model variables where literature supported an association with cervical cancer. We ran 6 different models with different main exposures. We adjusted simultaneously for the potential confounders: age (grouped), the period of the interview, marital status, place of residence and education level. We performed a test for heterogeneity on categorical variables and a score test for trends in the odds ratios on ordinal categorical variables using unconditional logistic regression Table 2. We used a Pearson correlation matrix to test for multicollinearity between independent variables in each model [21]. We used the Mantel-Haesnzel Chi-square test to assess differences in the prevalence of cofactors across different cancer types in different sets of controls Table 3.

## Population attributable fractions

We calculated population attributable fraction (PAF) using Miettinen's method given by $PAF = P_{ei}\left(\frac{OR_{adj}-1}{OR_{adj}}\right)$ [22]; where $P_{ei}$ is the proportion of cases exposed to a particular cofactor and $i$ is the exposure level. When a variable had three categories, we added the PAF for the categories. A PAF % represents the proportion of the disease attributable to a particular exposure. We then ranked the cofactors which were significant based on their PAF. All the statistical tests were done using STATA software version 16.0 (Stata Corp, college station, Tx). Statistical significance was considered at a two-sided α-level of 0.05.

## Results

Controls were relatively older compared to cases (Median age 50 years (IQR: 42–57) versus 48 years (IQR: 41–55)). The highest percentage of participants were in the age group of 45–54 years: 36.4% among cervical cancer cases and 33.8% among controls. Most of the study participants (cases = 84.2% and controls = 90.4%) lived in urban areas. More than half of the participants (58.5%) achieved a secondary or greater level of education Table 1.

Cases had nearly three times the risk of being HIV positive compared to controls ($OR_{adj}$ 2.83, 95% CI: 2.54–3.15). After controlling for the common set of confounders, a strong trend of elevated risk with higher parity was observed, relative to 1–2 children, with odds ratios of

**Table 1. Demographics: Cervical cancer cases and infection unrelated cancer control participants from the JCS.**

| Characteristics | Total (N = 9249 (100%)) N (%) | Cases (N = 3540 (100%)) n (%) | Controls (N = 5709 (100%)) n (%) |
|---|---|---|---|
| **Cervical cancer types** | | | |
| SCC | | 3364 (95.0) | |
| Adenocarcinoma | | 176 (5.0) | |
| **Demographics** | | | |
| **Age** | | | |
| Median (IQR) | 49 (42–56) | 48 (41–55) | 50 (42–57) |
| **Age** | | | |
| 25–34 | 795 (8.6) | 286 (8.1) | 525 (8.9) |
| 35–44 | 2410 (26.1) | 1010 (28.5) | 1400 (24.5) |
| 45–54 | 3220 (34.8) | 1288 (36.4) | 1932 (33.8) |
| 55–64 | 2824(30.5) | 956 (27.1) | 1868 (32.7) |
| **Period of interview** | | | |
| 1995–1999 | 1619 (17.5) | 766 (21.6) | 853 (14.8) |
| 2000–2004 | 1455 (15.7) | 477 (13.5) | 978 (17.1) |
| 2005–2009 | 2544 (27.5) | 994 (28.1) | 1550 (27.2) |
| 2010–2016 | 3631 (39.3) | 1303 (36.8) | 2328 (40.8) |
| **Marital Status** | | | |
| Never Married | 2224 (24.1) | 861 (24.3) | 1363 (23.9) |
| Married | 4152 (44.8) | 1581 (44.7) | 2571 (45.0) |
| Ever married | 2848 (30.8) | 1088 (30.7) | 1760 (30.8) |
| Missing data | 25 (0.3) | 10 (0.3) | 15 (0.3) |
| **Place of Residence** | | | |
| Rural | 1069 (11.5) | 546 (15.4) | 524 (9.1) |
| Urban | 8181 (88.1) | 2982 (84.2) | 5190 (90.4) |
| Missing data | 39 (0.4) | 13 (0.4) | 26 (0.5) |
| **Education Level** | | | |
| None | 1037 (11.2) | 515 (14.6) | 522 (9.1) |
| Primary | 2772 (30.0) | 1202 (34.0) | 1570 (27.5) |
| Secondary | 4930 (53.3) | 1707 (48.2) | 3223 (56.5) |
| Tertiary | 479 (5.2) | 97 (2.7) | 379 (6.6) |
| Missing data | 34 (0.4) | 19 (0.5) | 15 (0.3) |

Cancer Controls are unrelated to infection, Ever married includes widowed and divorced, IQR = Interquartile range, SCC = Squamous Cell Carcinoma, JCS = Johannesburg Cancer Study

Table 2. Multivariable analysis of lifestyle risk factors and cervical cancer participants from the JCS.

| Characteristics | Total (N = 9249) N (%) | Cases (N = 3540) n (%) | Controls (N = 5709) n (%) | Unadjusted OR (95%CI) | Adjusted Odds Ratios (95% CI) using appropriately chosen controls | Case control comparison p-value |
|---|---|---|---|---|---|---|
| Demographic factors | | | | | | |
| Age | | | | | | |
| 25–34 | 795 (8.6) | 286 (8.1) | 525 (8.9) | 1.10 (0.93–1.29) | 1.00 (0.96–1.37) | 0.906 |
| 35–44 | 2410 (26.1) | 1010 (28.5) | 1400 (24.5) | 1.41 (1.26–1.58) | **1.34 (1.17–1.52)** | **<0.001** |
| 45–54 | 3220 (34.8) | 1288 (36.4) | 1932 (33.8) | 1.30 (1.17–1.45) | **1.28 (1.14–1.43)** | **<0.001** |
| 55–64 | 2824(30.5) | 956 (27.1) | 1868 (32.7) | 1.00 | 1.00 | |
| p-value (trend) | | | | <0.001 | 0.016 | |
| Period of interview | | | | | | |
| 1995–1999 | 1619 (17.5) | 766 (21.6) | 853 (14.8) | 1.00 | 1.00 | |
| 2000–2004 | 1455 (15.7) | 477 (13.5) | 978 (17.1) | 0.54 (0.47–0.63) | **0.53 (0.45–0.62)** | **<0.001** |
| 2005–2009 | 2544 (27.5) | 994 (28.1) | 1550 (27.2) | 0.71 (0.63–0.81) | **0.67 (0.59–0.77)** | **<0.001** |
| 2010–2016 | 3631 (39.3) | 1303 (36.8) | 2328 (40.8) | 0.62 (0.55–0.70) | **0.59 (0.51–0.68)** | **<0.001** |
| p-value (trend) | | | | <0.001 | <0.001 | |
| Marital Status | | | | | | |
| Never Married | 2224 (24.1) | 861 (24.3) | 1363 (23.9) | 1.00 | 1.00 | |
| Married | 4152 (44.8) | 1581 (44.7) | 2571 (45.0) | 0.99 (0.87–1.12) | **1.14 (1.01–1.28)** | **0.023** |
| Ever married | 2848 (30.8) | 1088 (30.7) | 1760 (30.8) | 1.00 (0.87–1.14) | 1.11 (0.98–1.27) | 0.061 |
| Missing data | 25 (0.3) | 10 (0.3) | 15 (0.3) | - | - | |
| p-value (heterogeneity) | | | | 0.880 | 0.156 | |
| Place of residence | | | | | | |
| Rural | 1069 (11.5) | 546 (15.4) | 524 (9.1) | 1.85 (1.62–2.01) | **1.62 (1.41–1.86)** | **<0.001** |
| Urban | 8181 (88.1) | 2982 (84.2) | 5190 (90.4) | 1.00 | 1.00 | |
| Missing data | 39 (0.4) | 13 (0.4) | 26 (0.5) | - | - | |
| p-value (heterogeneity) | | | | <0.001 | <0.001 | |
| Education Level | | | | | | |
| None | 1037 (11.2) | 515 (14.6) | 522 (9.1) | 1.97 (1.72–2.25) | **2.01 (1.73–2.34)** | **<0.001** |
| Primary | 2772 (30.0) | 1202 (34.0) | 1570 (27.5) | 1.53 (1.39–1.68) | **1.60 (1.44–1.77)** | **<0.001** |
| Secondary and above | 5406 (58.5) | 1804 (51.0) | 3223 (63.1) | 1.00 | 1.00 | |
| Missing data | 34 (0.4) | 19 (0.5) | 15 (0.3) | | | |
| p-value (heterogeneity) | | | | <0.001 | <0.001 | |
| Lifestyle factors | | | | | | |
| HIV Status | | | | | **Model 1** | |
| Negative | 6759 (72.8) | 2217 (62.6) | 4542 (79.0) | 1.00 | 1.00 | |
| Positive | 2550 (27.2) | 1323 (37.4) | 1207 (21.0) | 2.32 (2.10–2.54) | **2.83 (2.53–3.14)** | **<0.001** |
| Missing data | - | - | - | | - | |
| p-value (heterogeneity) | | | | <0.001 | <0.001 | |
| Parity | | | | | **Model 2** | |
| 1–2 | 1595 (33.2) | 1138 (32.2) | 457 (35.5) | 1.00 | 1.00 | |
| 3+ | 3021 (668.9) | 2288 (65.0) | 733 (57.0) | 1.49 (1.36–1.63) | **1.25 (1.07–1.46)** | **0.005** |
| Missing data | 190 (4.0) | 94 (2.7) | 96 (7.5) | 0.65 (0.50–0.83) | **0.57 (0.44–0.74)** | **0.002** |
| p-value (trend) | | | | <0.001 | 0.002 | |
| Smoking | | | | | **Model 3** | |

*(Continued)*

**Table 2.** (Continued)

| Characteristics | Total (N = 9249) N (%) | Cases (N = 3540) n (%) | Controls (N = 5709) n (%) | Unadjusted OR (95%CI) | Adjusted Odds Ratios (95% CI) using appropriately chosen controls | Case control comparison |
|---|---|---|---|---|---|---|
| | | | | | | p-value |
| Never | 6737(83.3) | 2830 (79.9) | 3807 (85.9) | 1.00 | 1.00 | |
| Ex-Smoker | 424 (5.3) | 209 (5.9) | 215 (4.8) | 1.31 (1.07–1.59) | **1.26 (1.02–1.55)** | **0.030** |
| Current | 901 (11.3) | 498 (14.1) | 403 (9.2) | 1.66 (1.44–1.91) | **1.55 (1.34–1.80)** | **<0.001** |
| Missing data | 8 (0.1) | 3 (0.1) | 5 (0.1) | - | - | |
| p-value (heterogeneity) | | | | <0.001 | <0.001 | |
| Alcohol consumption | | | | | **Model 4** | |
| Never | 3538 (79.3) | 2781 (78.6) | 757 (82.5) | 1.00 | 1.00 | |
| Light | 243 (5.5) | 194 (5.5) | 49 (5.3) | 1.08 (0.77–1.49) | 1.20 (0.86–1.68) | 0.287 |
| Heavy | 677 (15.2) | 565 (16.0) | 112 (12.2) | 1.37 (1.10–1.71) | **1.44 (1.15–1.81)** | **0.002** |
| Missing data | - | - | - | - | - | |
| p-value (heterogeneity) | | | | 0.017 | 0.078 | |
| Contraceptive use | | | | | **Model 5** | |
| Never (injectable and oral) | 1833 (38.2) | 1262 (35.7) | 571 (45.3) | 1.00 | 1.00 | |
| Ever Oral/ not injectable | 578 (12.0) | 408 (11.5) | 170 (13.5) | 1.09 (0.89–1.33) | 1.09 (0.88–1.36) | 0.847 |
| Ever Injectable/ not oral | 1526 (31.8) | 1226 (34.5) | 305 (24.2) | 1.81 (1.54–2.13) | **1.34 (1.11–1.61)** | **0.002** |
| Ever oral and Injectable | 844 (17.6) | 637 (18.0) | 207 (16.4) | 1.39 (1.16–1.68) | 1.22 (1.00–1.50) | 0.366 |
| Ever injectable and/ or oral | 2948 (61.4) | 2266 (64.0) | 709 (54.5) | 1.50 (1.32–1.71) | **1.17 (1.01–1.37)** | **0.039** |
| Missing data | 19 (0.4) | 12 (0.3) | 7 (0.6) | - | - | |
| p-value (heterogeneity) | | | | <0.001 | 0.183 | |
| Number of sexual partners | | | | | **Model 6** | |
| 0–1 | 661 (7.2) | 233 (6.6) | 428 (7.5) | 1.00 | 1.00 | |
| 2–5 | 5384 (58.0) | 2123 (60.0) | 3243 (56.8) | 1.20 (1.01–1.42) | 1.15 (0.96–1.37) | 0.129 |
| 6+ | 1007 (10.9) | 397 (11.2) | 610 (10.7) | 1.20 (0.98–1.47) | 1.11 (0.89–1.38) | 0.298 |
| Missing data | 2215 (24.0) | 787 (22.2) | 1428 (25.0) | 1.01 (0.84–1.21) | 1.09 (0.88–1.34) | 0.378 |
| p-value (trend) | | | | 0.040 | 0.020 | |

Notes

a. The total for alcohol, smoking and contraceptive do not add up to the whole total because some of the cancers were removed from the list of controls (see Table 3 in the appedix).

b. Odds Ratios were adjusted for age, education level, marital status, period of the interview, place of residence (rural and urban).

c. The sets of controls that were used are different in each model depending on the association between the exposure and cervical cancer.

1.25 (95% CI: 1.07–1.46) for women reporting three or more children Table 2. Compared to never smokers, the risk of cervical cancer significantly increased among women who were current smokers (OR$_{adj}$ 1.55, 95% CI: 1.34–1.80) and ex-smokers (OR$_{adj}$ 1.26, 95% CI: 1.02–1.55). The odds ratios for cervical cancer were 1.44 (95% CI: 1.26–1.65) for women in the age group 35–44 years and 1.33 (95% CI: 1.18–1.50) for those in the age group 45–54 years old compared to women in the age group 55–64 years. The risk of cervical cancer increased among married women compared to never married (OR$_{adj}$ 1.14, 95% CI: 1.01–1.28). The risk of cervical cancer

**Table 3. Age-adjusted p-value for hetrogeneity in prevalence of cofactors across different cancer types in controls.**

| Co-factors | Infection unrelated cancer controls | | | Smoking unrelated cancer controls | | | Alcohol unrelated cancer controls | | | Hormonal contraceptive unrelated cancer controls | | |
|---|---|---|---|---|---|---|---|---|---|---|---|---|
| | *DF | Chi- square | p-value | DF | Chi-square | #p-value | DF | Chi-square | p-value | DF | Chi-square | p-value |
| Parity | - | - | - | - | - | - | - | - | - | 10 | 9.02 | 0.417 |
| HIV | 9 | 15.2 | 0.076 | - | - | - | - | - | - | - | - | - |
| Number sexual partner | 9 | 11.02 | 0.274 | - | - | - | - | - | - | - | - | - |
| Smoking | - | - | - | 3 | 5.44 | 0.142 | - | - | - | - | - | - |
| Alcohol use | - | - | - | - | - | - | 8 | 17.13 | 0.029 | - | - | - |
| Contraceptive use | - | - | - | - | - | - | - | - | - | 15 | 30.82 | 0.051 |

*DF = degrees of freedom

#p-value <0.05

among women who reported having consumed at least 47.1 grams of alcohol per week (heavy alcohol consumption) was 44% higher ($OR_{adj}$ 1.44, 95% CI: 1.15–1.81) compared to women who reported having never consumed alcohol. Cases were more likely than controls to have ever used injectable contraceptives ($OR_{adj}$ 1.34, 95% CI: 1.11–1.61) and ever used oral or injectable contraceptives ($OR_{adj}$ 1.17, 95% CI: 1.01–1.37). A lower level of educational attainment was associated with an increased risk of cervical cancer: Education (None versus Secondary and above $OR_{adj}$ 2.01, 95% CI: 1.73–2.34) and (Primary versus Secondary and above $OR_{ad}$ 1.60, 95% CI: 1.44–1.77). Living in rural compared to urban areas increased the risk of cervical cancer by 62% ($OR_{adj}$ 1.62, 95% CI: 1.44–1.77).

Our estimated PAF of cervical cancer for the 7 modifiable cofactors ranged from HIV(positive) (17.6%), lower education attainment (no education or primary) (16.9%), higher parity (3+ children) (12.6%), contraceptive use (Ever injectable and/ or oral) (8.9%), alcohol consumption (light and heavy) (5.6%), smoking (ex-smoker and current smoker) (5.1%) and 4.4% for the place of residence (rural) Fig 2. The number of sexual partners were excluded as it was not statistically significant.

We tested the robustness of our control selections on the assumption that the prevalence of exposure for each of the cancer types chosen in each of the comparisons should be homogeneous. We therefore, calculated an adjusted (age, number of sexual partners and education level) Mantel-Haenszel Chi-square tests of heterogeneity for each type of comparisons. We observed no differences (p> 0.05) in the prevalence of cofactors across different cancer types in the three control arms except for alcohol Table 3.

## Discussion

In descending order, cervical cancer was associated with being HIV-positive, educational attainment, higher parity, contraceptive use, heavy consumption of alcohol, current smoking and residing in rural areas among black South African women.

In the current study, cervical cancer risk was significantly associated with HIV infection, which was ranked as the most important risk factor, with a PAF of 17.6%. This implies that about 18% of cervical cancers would be prevented if individuals were not infected with HIV. This is particularly important in South Africa where black women are disproportionally affected by HIV compared to other racial groups [23]. The association between HIV and cervical cancer is supported by several studies in low- and middle-income countries [24, 25], showing that during HPV pathogenesis, coinfection with HIV increases HPV viral persistence [25]. Our finding is similar to the previous study on the spectrum of HIV-1 related cancers in South Africa which used the JCS data. In that study, the risk of cervical cancer was elevated in HIV-1 positive women, OR = 1.6 (95% CI: 1.1–2.3) [26]. Similarly, in a Ugandan study, HIV positive

women had an increased risk for cervical cancer ($OR_{adj}$ = 2.4, 95% CI: 1.1–4.4) [8]. Efforts to reduce new HIV infections among black women in South Africa would likely reduce the incidence of cervical cancer in these women in future.

High incidence rates of cervical cancer have been reported among women with low socio-economic status in both high and low-income settings [27, 28]. Socio-economic status may affect cervical cancer incidence via levels of educational attainment. Education, a measure of socio-economic status, is inversely related to cervical cancer risk [29]. In a study from Spain and Columbia having no education was associated with cervical cancer (OR = 2.5, 95% CI: 1.6–3.9) [30]. Similarly, our study found having no education or completing primary school was associated with an increased risk of cervical cancer. Our results indicate that 16.9% of the cervical cancers would have been prevented if women with primary education had secondary and above education. Future investments in education should drive cervical cancer rates downwards.

In our study, higher parity (was possibly related with age at first sexual debut and possibly early exposure to HPV infection) was associated with the risk of developing cervical cancer and a PAF of 12.6%. Our findings are in agreement with Briton et al. [31], Castellsague et al. [5] and Jensen et al. [32] who demonstrated an association between higher parity (2 or more children) and cervical cancer risk. The International Agency for Research on Cancer (IARC) showed a 3.8 (95% CI: 2.7–5.5) times risk for cervical cancer with 7 or more full-term pregnancies [33]. Having 2 or more children was associated with persistent HPV infection which facilitated the development of cervical cancer [32]. Similar to our study, PAF's of between 24% and 44% were reported for the USA and Italy [34] and 42% in Costa Rica for higher parity [10]. Evidence suggests that the burden of cervical cancer should decrease if average family sizes decrease [33, 35].

Prolonged use of hormonal contraceptive has been associated with cervical cancer [36]. A previous study from the JCS demonstrated an association between the duration of hormonal contraceptive use and cervical cancer [13]. In our study, we did not consider the duration of contraceptive use and cervical cancer. Nonetheless, we reported OR = 1.48 (95% CI: 1.24–1.77) and a PAF of 8.9%. Our findings which showed an association between the use of both injectable and oral contraceptive use and cervical cancer are broadly similar to the JCS findings of Urban et al. [13], (OR = 1.38, 95% CI: 1.08–1.77), as well as for injectable only and cervical cancer (OR = 1.58, 95% CI: 1.16–2.15). Our finding is also comparable to that of Appleby et al. [36], which contains earlier JCS data, where combined contraceptive use was associated with cervical cancer. Similarly, in Latin America, Herrero et al. [37] found an association between injectable contraceptive use and cervical cancer, which reduces significantly after cessation of use. Thus, prolonged use of hormonal contraceptives should be time-limited to avoid excess risks.

The evidence concerning alcohol consumption and the risk for cervical cancer is equivocal. Some studies demonstrated an association with alcohol consumption [9, 38, 39] but others did not [40, 41]. Our study, however, found a significantly elevated risk for cervical cancer in heavy drinkers. Our data on PAF suggest that 5.6% of cervical cancer cases would be avoided if alcohol consumption were reduced. It is possible that women who were heavy drinkers in our study may have also been involved in other high-risk behaviours such as smoking, having multiple sexual partners and other behaviours that promote the acquisition of hr-HPV [42]. Notably, lack of enough data on cervical screening made it difficult to control for it, thus we can not preclude the effects of residual confounding.

Current smoking has consistently been associated with cervical cancer pathogenesis. We reported an OR of 1.55 (95% CI: 1.34–1.80) for current smokers and a PAF of 5.1% for smoking. A 2008 JCS study, found current smokers in South Africa had an increased risk of cervical

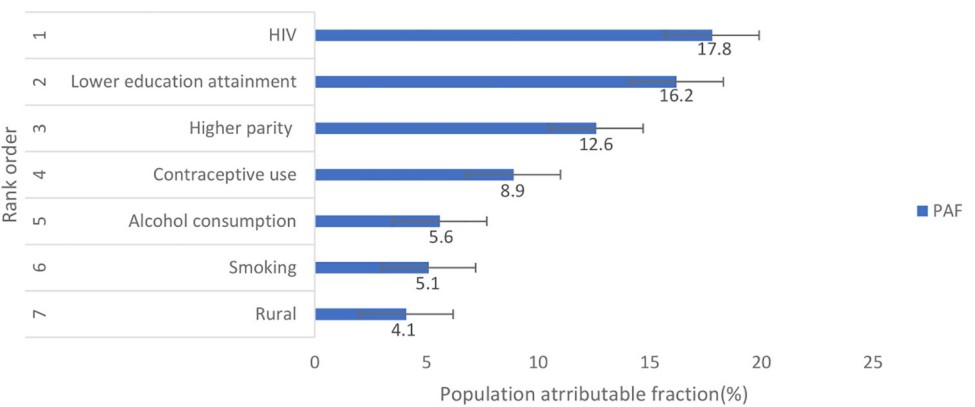

**Fig 2. Population attributable fraction of the cofactors on cervical cancer.**

cancers relative to never smokers (OR = 1.5, 95% CI: 1.2–1.8) [43]. Similarly, Roula et al. [44] in a European prospective cohort study, showed current smokers had a Hazard Ratio (HR) of 1.6 (95% CI: 1.4–2.5) for cervical cancer. Besides, a 2011 international study by IARC showed that current smokers compared to never smokers had a OR = 1.94 (95% CI: 1.26–2.98) risk of developing cervical cancer [45]. While smoking is an important risk factor for cervical in other populations, it might not be as important a risk factor in black South African women, where the prevalence of smoking is low (4.1%) [46]. Continuing progressive anti-smoking laws and health promotion in South Africa is needed to retain low smoking prevalence among Black females.

We found an association between residing in the rural area and cervical cancer. Our finding is comparable to the findings of Yang et al. [47], which showed an association between rural area and cervical cancer (OR = 1.46, 95% CI: 1.03–2.06) and a PAF of 4.4%. This could be due to a lack of cervical cancer screening services, poverty in rural areas, lower education attainment and low uptake in cervical cancer screening. Vhuromu et al. [48], reported low utilization of cervical cancer screening in rural areas. Improved health systems and new methods of cervical cancer screening services in rural areas may result in an improved uptake in cervical cancer screening. Such efforts would likely reduce cervical cancer cases in rural areas.

Sexual intercourse is the main route for HPV transmission. Having multiple sexual partners increases the risk of HPV infection among women which may result in cervical cancer. Other studies have demonstrated an association between the number of sexual partners and cervical cancer risk [49, 50]. Liu et al. [49], used a meta-analysis of 41 studies and found the number of sexual partners was an independent risk factor for cervical cancer even after adjusting for HPV infection. Appleby et al. [50], in an international collaboration of 21 epidemiological studies demonstrated an association between the number of sexual partners and cervical cancer. In our study, we did not find an association between the number of sexual partners and the risk of cervical cancer. However, there was a small increase in risk which was not significant Table 2. The possible explanation for non signifant finding in our study could be a result of small samples size among women aged 25–34, reporting bias (erroneously reporting fewer

sexual partners than in reality), confounding by HIV, and no partner data available on male partners.

Our study supports the importance given to the cofactors of cervical cancer and the need for effective interventions of these cofactors. In South Africa, cervical cancer rates could increase or decrease depending on the effectiveness of public health policies. The PAF takes into account the prevalence of the cofactor and adjusted odds ratios, underscoring that an increase in the prevalence of these cofactors could have a notable effect on cervical cancer incidence among black women in South Africa.

There are some strengths to our study. The larger sample size allowed us to estimate the contribution of key lifestyle cofactors more reliably than before. The selection of appropriate sets of cancer controls unrelated to a specific exposure of interest minimised the bias of the estimates related to the exposure, and referral, interviewer and recall biases [17, 18]. Both cases and controls were patients with cancers and they are likely to remember their past exposures in a similar way. This is the first study that attempts to measure the relative importance of a wide range of risk factors for cervical cancer in the same study population, by adjusting for a common set of confounders.

This study has several limitations. The effects obtained were not adjusted for frequency of Pap smear screening since the data available was self-reported and the question of Pap smear was only added in 2001. We could not control for unmeasured confounders despite adjusting for known confounders such as age and HIV status. Since the study only focused on black South African women, mainly resident in Johannesburg and Soweto, the findings from this study may not be generalizable to the entire population of South Africa. Another limitation of this case-control study is reverse causality whereby people with long-standing conditions may change their lifestyle. We mitigated this by reclassifying those who reported quitting smoking five years before diagnosis as current smokers, in case they altered their habits because of underlying health problems. Although the questions on alcohol consumption were focused on before the participant became ill, we could not perform a similar re-classification as for smoking. Because different exposures required different control selections, we avoided measuring combined exposures e.g. smoking and drinking.

In conclusion, in order of importance, HIV-positivity, educational attainment, parity, hormonal contraceptive use, alcohol, smoking and residing in rural area were associated with cervical cancer among black South African women. These women should be prioritized in opportunistic or planned cervical cancer screening programs. Our findings confirm previosuly known cofactors of cervical cancer and provide a rank order of risks that could be used locally to target educational messaging and appropriate interventions.

## Supporting information

**S1 Table. List of cancers included in each control group.**
(DOCX)

**S1 Data.**
(XLS)

## Acknowledgments

We acknoweldge all who provided feedback to improve the manuscript.

## Author Contributions

**Conceptualization:** Mwiza Gideon Singini.

**Data curation:** Wenlong Carl Chen, Melitah Motlhale, Abram Bunya Kamiza, Mazvita Muchengeti.

**Formal analysis:** Mwiza Gideon Singini.

**Funding acquisition:** Freddy Sitas, Debbie Bradshaw, Cathryn M. Lewis, Christopher G. Mathew, Tim Waterboer, Robert Newton.

**Investigation:** Debbie Bradshaw, Christopher G. Mathew, Tim Waterboer, Robert Newton, Elvira Singh.

**Methodology:** Mwiza Gideon Singini.

**Software:** Mwiza Gideon Singini.

**Supervision:** Freddy Sitas, Elvira Singh.

**Visualization:** Wenlong Carl Chen.

**Writing – original draft:** Mwiza Gideon Singini.

**Writing – review & editing:** Mwiza Gideon Singini, Freddy Sitas, Debbie Bradshaw, Wenlong Carl Chen, Melitah Motlhale, Abram Bunya Kamiza, Chantal Babb de Villiers, Cathryn M. Lewis, Christopher G. Mathew, Tim Waterboer, Robert Newton, Mazvita Muchengeti, Elvira Singh.

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
