## [Decision Letter · Decision Letter 0]

10 Sep 2021

PONE-D-21-18984Ranking lifestyle risk factors for cervical cancer among Black women: A Case-Control study from Johannesburg, South AfricaPLOS ONE

Dear Dr. Singh,

Thank you for submitting your manuscript to PLOS ONE. After careful consideration, we feel that it has merit but does not fully meet PLOS ONE’s publication criteria as it currently stands. Therefore, we invite you to submit a revised version of the manuscript that addresses the points raised during the review process. Please submit your revised manuscript by Oct 25 2021 11:59PM. If you will need more time than this to complete your revisions, please reply to this message or contact the journal office at plosone@plos.org. Please include the following items when submitting your revised manuscript:A rebuttal letter that responds to each point raised by the academic editor and reviewer(s). You should upload this letter as a separate file labeled 'Response to Reviewers'.A marked-up copy of your manuscript that highlights changes made to the original version. You should upload this as a separate file labeled 'Revised Manuscript with Track Changes'.An unmarked version of your revised paper without tracked changes. You should upload this as a separate file labeled 'Manuscript'.

We look forward to receiving your revised manuscript.

Kind regards,

Nülüfer Erbil, Ph.D, Prof.

Academic Editor

PLOS ONE

2. Please include additional information regarding the survey or questionnaire used in the study and ensure that you have provided sufficient details that others could replicate the analyses. For instance, if you developed the survey or questionnaire as part of this study and it is not under a copyright more restrictive than CC-BY, please include a copy, in both the original language and English, as Supporting Information. If the questionnaire is published, please provide a citation to the (1) questionnaire and/or (2) original publication associated with the questionnaire.

3. In your ethics statement in the Methods section and in the online submission form, please provide additional information about the data used in your retrospective study. Specifically, please ensure that you have discussed whether all data were fully anonymized before you accessed them and/or whether the IRB or ethics committee waived the requirement for informed consent. If patients provided informed written consent to have data from their medical records used in research, please include this information.

4. Please include your actual numerical p-values in Table 2

Additional Editor Comments (if provided):

Reviewers' comments:

Reviewer's Responses to Questions

**Comments to the Author**

1. Is the manuscript technically sound, and do the data support the conclusions?

Reviewer #1: Yes

Reviewer #2: Yes

2. Has the statistical analysis been performed appropriately and rigorously? 

Reviewer #1: Yes

Reviewer #2: Yes

3. Have the authors made all data underlying the findings in their manuscript fully available?

Reviewer #1: Yes

Reviewer #2: Yes

4. Is the manuscript presented in an intelligible fashion and written in standard English?

Reviewer #1: Yes

Reviewer #2: Yes

5. Review Comments to the Author

Reviewer #1: Dear Authors,

Although the manuscript is well statistics, according to previous references, all information that we had known with the lifestyle risk factors for cervical cancer among Black women, like these related references that authors cited, so for more contribution, suggest authors clarify and mentation what these JCS data that the medical system or policy-related cervical cancer could improve. Thank you for your effort.

Reviewer #2: This is a good study to determine the risk factors for cervical cancer, one of the most common cancer types in women.It is a case-control study on a large sample that will contribute to the literature.

6. PLOS authors have the option to publish the peer review history of their article (what does this mean?). If published, this will include your full peer review and any attached files.

Reviewer #1: No

Reviewer #2: No

---

## [Author Response · Author response to Decision Letter 0]

27 Sep 2021

Response :

Thank you for your comment: The manuscript has been formatted and meets PLOSE ONE’s style requirement.

2. Please include additional information regarding the survey or questionnaire used in the study and ensure that you have provided sufficient details that others could replicate the analyses. For instance, if you developed the survey or questionnaire as part of this study and it is not under a copyright more restrictive than CC-BY, please include a copy, in both the original language and English, as Supporting Information. If the questionnaire is published, please provide a citation to the (1) questionnaire and/or (2) original publication associated with the questionnaire.

Response: 

Thank you for your comment: Additional information regarding the questionnaire has been added to the manuscript line 141-147. Page 5, as follows: 

The questionnaire included questions on the following: socio-demographic factors such as; place of birth and residence, marital status, education, the home language of parents. Enviromental exposures such as method of cooking and heating. Lifestyle factors such as; smoking by type of tobacco and amounts smoked, snuff (sniffed tobacco) use, alcohol consumption by type, parity, use of oral and injectable contraceptives, number of sexual partners. On occupations, self-reported use of Anti-Retroviral Therapy (ART) (since 2005), PAP smear (2001) and self-reported history of diabetes.

The questionnaire has also been included as a Supplementary material.

3. In your ethics statement in the Methods section and in the online submission form, please provide additional information about the data used in your retrospective study. Specifically, please ensure that you have discussed whether all data were fully anonymized before you accessed them and/or whether the IRB or ethics committee waived the requirement for informed consent. If patients provided informed written consent to have data from their medical records used in research, please include this information.

Response:

Thank you for your comment: Information on data anonymization has been discussed in the revised manuscript lines 147-152 page 5 as follows: 

The JCS and the current study were approved by the University of the Witwatersrand Human Research Ethics Committee (Medical) (certificate number for the current study. M200252). In the JCS, participants gave written informed or wittnessd consent to once-off interview and optional blood draw and to have their information and blood sample anonymized. Any future investigations require approval of the University of the Witwatersrand Human Research Ethics Committee HREC [16].

The information has also been included in the online submission form.

4. Please include your actual numerical p-values in Table 2.

Response:

Thank you for your comment: The actual numerical p-values in Table 2 have been added. Pages 13-15

Response:

Data cannot be shared publicly because of ethics policy at University of Witwatersrand, whereby any new analyses require Human Research Ethics Committee approval. Data are available from the SA-NCR /National Health Laboratory Services. (contact : elviras@nicd.ac.za) for researchers who meet the relevant ethics criteria for access to these data.

Reviewer 1:

Although the manuscript is well statistics, according to previous references, all information that we had known with the lifestyle risk factors for cervical cancer among Black women, like these related references that authors cited, so for more contribution, suggest authors clarify and mentation what these JCS data that the medical system or policy-related cervical cancer could improve. Thank you for your effort.

Response:

Thank you very much for your question:

Our findings have important policy implications. Firstly, ranking of lifestyle risk factors for cervical cancer can inform medical systems on which key risk factors to integrate in cervical cancer education programmes. This would help health care personnel that are involved in advising women about cervical cancer screening, to understand the public-health gains that result by minimizing the risk of each factor. Second, most of the previous studies were conducted in women of European ancestry. We have amended the concluding sentence as follows:

In conclusion, in order of importance, HIV-positivity, educational attainment, parity, hormonal contraceptive use, alcohol, smoking and residing in rural area were associated with cervical cancer among black South African women. These women should be prioritized in opportunistic or planned cervical cancer screening programs. Our findings confirm previosuly known cofactors of cervical cancer and provide a rank order of risks that could be used locally to target educational messaging and appropriate interventions.

Reviewer #2: 

This is a good study to determine the risk factors for cervical cancer, one of the most common cancer types in women. It is a case-control study on a large sample that will contribute to the literature. 

We thank the reviewer for their comments.

---

## [Decision Letter · Decision Letter 1]

22 Oct 2021

PONE-D-21-18984R1Ranking lifestyle risk factors for cervical cancer among Black women: A Case-Control study from Johannesburg, South AfricaPLOS ONE

Dear Dr. Singh,

Thank you for submitting your manuscript to PLOS ONE. After careful consideration, we feel that it has merit but does not fully meet PLOS ONE’s publication criteria as it currently stands. Therefore, we invite you to submit a revised version of the manuscript that addresses the points raised during the review process.

We look forward to receiving your revised manuscript.

Kind regards,

Nülüfer Erbil, Ph.D, Prof.

Academic Editor

PLOS ONE

Journal Requirements:

Reviewers' comments:

Reviewer's Responses to Questions

**Comments to the Author**

1. If the authors have adequately addressed your comments raised in a previous round of review and you feel that this manuscript is now acceptable for publication, you may indicate that here to bypass the “Comments to the Author” section, enter your conflict of interest statement in the “Confidential to Editor” section, and submit your "Accept" recommendation.

Reviewer #1: (No Response)

Reviewer #2: All comments have been addressed

2. Is the manuscript technically sound, and do the data support the conclusions?

Reviewer #1: Yes

Reviewer #2: (No Response)

3. Has the statistical analysis been performed appropriately and rigorously? 

Reviewer #1: Yes

Reviewer #2: Yes

4. Have the authors made all data underlying the findings in their manuscript fully available?

Reviewer #1: Yes

Reviewer #2: Yes

5. Is the manuscript presented in an intelligible fashion and written in standard English?

Reviewer #1: Yes

Reviewer #2: Yes

6. Review Comments to the Author

Reviewer #1: Dear authors,

The revised manuscript is better, but Result section, “A total of, 26,263 participants were enrolled in the JCS between 1995 to 2016. A total of 9,249 women were included in this study of whom 3,540 (38.3%) had invasive cervical cancer (cases) and 5,709 were designated as controls, (61.7%) as described in Figure 1.”The figure is vague, not easy to read follow the authors’ context, please clarify.

Thank you.

Reviewer #2: An original study done in a special group. It is a good work that will contribute to the literature.

7. PLOS authors have the option to publish the peer review history of their article (what does this mean?). If published, this will include your full peer review and any attached files.

Reviewer #1: No

Reviewer #2: No

---

## [Author Response · Author response to Decision Letter 1]

29 Oct 2021

Please review your reference list to ensure that it is complete and correct. 

Response:

Reference list has been reviewed, it is complete and correct 

Reviewer #1: Dear authors,

The revised manuscript is better, but Result section, “A total of, 26,263 participants were enrolled in the JCS between 1995 to 2016. A total of 9,249 women were included in this study of whom 3,540 (38.3%) had invasive cervical cancer (cases) and 5,709 were designated as controls, (61.7%) as described in Figure 1.”The figure is vague, not easy to read follow the authors’ context, please clarify.

Thank you.

Response:

Thank you so much for your question:

Figure 1 has been revised for easy to read. The information on the results section regarding figure 1, has been placed in the methods section where the description of exclusion was done. (lines 161-184 of the manuscript).

A total of, 26,263 participants were enrolled in the JCS between 1995 to 2016. We excluded 8,677 (33.0%) males, 3,105 (17.6%) women older than 65 years or younger than 25 years, 663 (4.6%) non-cancer participants, 945(6.9%) non-South Africans, 1 with missing data, 640 (5.0%) with missing HIV-status and 691 (5.7%) with primary site unknown malignancy. From those with cancer of the cervix, we excluded International Classification of Diseases for Oncology (ICD-O) codes such as ( ICD-O morphology: 8010-8050 epithelial neoplasm (n=98, 2.6%), (ICD-O morphology: 8000 and 8001) not otherwise specified (n=9, 0.2%), (ICD-O morphology: 8560, 8570 and 8574) complex epithelial neoplasms (n=108, 2.8%) and other minor histological types (ICD-O morphology: 8098,8170, 8200, 8272, 8310, 8441,8460, 8480-8490, 8500,8650 and 8931-9100) (n=64, 1.7%) Figure 1.

Figure 1 also outlines the criteria used to select cases and controls. The control groups were arranged into four sets. Each set comprised of women diagnosed with different cancer types that are unrelated to the exposures of interest (these being infection, smoking, alcohol, parity, number of sexual partners and use of hormonal contraceptives) [16–18] Figure 1. Cancer controls unrelated to infection constituted the highest number of controls S1 Table. The demographic analyses excluded 2,009 (26.0%) women with cancers related to infections. Analyses relating to smoking excluded 2,218 (28.7%) women with cancers related to smoking and infections. Analyses relating to alcohol excluded 6,800 (60.4%) women with cancers related to alcohol and infections. Analyses relating to hormonal contraceptives excluded 6,458 (57.4%) women with cancers related to hormonal contraceptives and infections. Of the remaining 9,249 women, 3,540 (38.3%) had invasive cervical cancer (defined as cases) and 5,709 (61.7%) had other cancers (defined as controls) that were not related to the exposure of interest Figure1.

Reviewer #2: 

An original study done in a special group. It is a good work that will contribute to the literature.

We thank the reviewer for their comments.

---

## [Decision Letter · Decision Letter 2]

8 Nov 2021

Ranking lifestyle risk factors for cervical cancer among Black women: A Case-Control study from Johannesburg, South Africa

PONE-D-21-18984R2

Dear Dr. Singh,

We’re pleased to inform you that your manuscript has been judged scientifically suitable for publication and will be formally accepted for publication once it meets all outstanding technical requirements.

Kind regards,

Nülüfer Erbil, Ph.D, Prof.

Academic Editor

PLOS ONE

Additional Editor Comments (optional):

Reviewers' comments:

Reviewer's Responses to Questions

**Comments to the Author**

1. If the authors have adequately addressed your comments raised in a previous round of review and you feel that this manuscript is now acceptable for publication, you may indicate that here to bypass the “Comments to the Author” section, enter your conflict of interest statement in the “Confidential to Editor” section, and submit your "Accept" recommendation.

Reviewer #1: (No Response)

2. Is the manuscript technically sound, and do the data support the conclusions?

Reviewer #1: (No Response)

3. Has the statistical analysis been performed appropriately and rigorously? 

Reviewer #1: (No Response)

4. Have the authors made all data underlying the findings in their manuscript fully available?

Reviewer #1: (No Response)

5. Is the manuscript presented in an intelligible fashion and written in standard English?

Reviewer #1: (No Response)

6. Review Comments to the Author

Reviewer #1: DEAR Author,

The title with "Ranking lifestyle risk factors for cervical cancer among Black women: A Case-Control

study from Johannesburg, South Africa. " revised manuscript is much better. I have no other comments.

Thank you.

7. PLOS authors have the option to publish the peer review history of their article (what does this mean?). If published, this will include your full peer review and any attached files.

Reviewer #1: No

---

## [Editor Report · Acceptance letter]

17 Nov 2021

PONE-D-21-18984R2 

Ranking lifestyle risk factors for cervical cancer among Black women: A Case-Control study from Johannesburg, South Africa 

Dear Dr. Singh:

I'm pleased to inform you that your manuscript has been deemed suitable for publication in PLOS ONE. Congratulations! Your manuscript is now with our production department. 

Kind regards, 

on behalf of

Dr. Nülüfer Erbil 

Academic Editor

PLOS ONE